# The Brief Health Literacy Scale for Adults: Adaptation and Validation of the Health Literacy for School-Aged Children Questionnaire

**DOI:** 10.3390/ijerph20227071

**Published:** 2023-11-16

**Authors:** Stinne Eika Rasmussen, Anna Aaby, Anne Søjbjerg, Anna Mygind, Helle Terkildsen Maindal, Olli Paakkari, Kaj Sparle Christensen

**Affiliations:** 1Research Unit for General Practice, Bartholins Allé 2, 8000 Aarhus C, Denmark; annsoj@ph.au.dk (A.S.); mygind@ph.au.dk (A.M.); kasc@ph.au.dk (K.S.C.); 2Department of Public Health, Aarhus University, Bartholins Allé 2, 8000 Aarhus C, Denmark; annaab@rm.dk (A.A.); htm@ph.au.dk (H.T.M.); 3Faculty of Sport and Health Sciences, Research Centre for Health Promotion, University of Jyväskylä, Keskussairaalantie 4, 40014 Jyväskylä, Finland; olli.paakkari@jyu.fi

**Keywords:** health literacy, psychometrics, surveys and questionnaires, adults, health literacy for school-aged children, measuring instrument, Rasch analysis

## Abstract

The Health Literacy for School-Aged Children (HLSAC) is a brief, generic instrument measuring health literacy among school-aged children. Given its brevity and broad conceptualization of health literacy, the HLSAC is a potentially valuable measuring instrument among adults as well. This validation study aimed to adapt the HLSAC questionnaire to an adult population through assessment of content validity and subsequently determine the structural validity of the adapted instrument, the Brief Health Literacy scale for Adults (B-HLA). The content validity of the HLSAC was assessed through interviews with respondents and experts, and the structural validity of the adapted instrument (B-HLA) was evaluated using Rasch analysis. The content validity assessment (*n* = 25) gave rise to adjustments in the wording of five items. The B-HLA demonstrated an overall misfit to the Rasch model (*n* = 290). Items 6 and 8 had the poorest individual fits. We found no signs of local dependency or differential item functioning concerning sex, age, education, and native language. The B-HLA demonstrated unidimensionality and ability to discriminate across health literacy levels (PSI = 0.80). Discarding items 6 or 8 resulted in an overall model fit and individual fit of all items. In conclusion, the B-HLA appears to be a valid and reliable instrument for assessing health literacy among adults.

## 1. Introduction

Health literacy is a dynamic construct encompassing “the combination of personal competencies and situational resources needed for people to access, understand, appraise, and use information and services to make decisions about health” [1]. Health literacy is acknowledged as an important determinant of health. Inadequate health literacy is associated with poor health status [2,3,4], risky health behaviors [4,5], more and longer hospitalizations [2,6], and even increased mortality [2,7,8]. These associations suggest potential health effects of initiatives aimed at improving health literacy or reducing the negative impact of health literacy challenges. Furthermore, by recognizing the socioeconomic gradient in the distribution of health literacy, such initiatives may become key in mitigating social inequality in health [4,9]. Accordingly, research on novel health initiatives should consider health literacy as a relevant outcome or covariate characteristic of the target population.

Recent years have seen an exponential growth in the number of published papers about health literacy [10], and many papers state a broad definition of health literacy, like the above [11]. However, the instruments used to measure health literacy in population research frequently reflect a narrower concept of health literacy [10,12,13,14]. Furthermore, the available health literacy measuring instruments based upon a broader conceptualization of health literacy tend to be lengthy, which sometimes precludes their use due to practical reasons or leads to item attrition among respondents [15,16]. Particularly, the risk of missing data is likely to increase when the instruments are combined with other patient-reported outcome measures in research. Another prevalent challenge in relation to the application of health literacy measuring instruments is the frequent desire to report a sum score as an indicator of the health literacy level of the respondent or the population, which proves difficult since many instruments are not considered unidimensional, i.e., measuring only one single underlying construct [17,18,19,20,21,22,23,24]. Hence, a need exists for brief, generic, and unidimensional health literacy measuring instruments that conceptualize health literacy broadly.

The Health Literacy for School-Aged Children (HLSAC) questionnaire is a 10-item instrument measuring health literacy within five core components: theoretical knowledge, practical knowledge, critical thinking, self-awareness, and citizenship [25]. HLSAC is based on a broad understanding of health literacy comprising a range of knowledge and competencies that help people understand themselves, others, and the world in a way that will enable them to make sound health decisions [26] appropriate for their age and situation. The HLSAC thus measures the respondents’ currently experienced abilities in relation to the five components and does not refer to any age-specific health challenges or use age-specific cut-offs. The HLSAC was originally developed to inform the integration of health literacy in school curricula and has been validated among children and adolescents in several countries, including Denmark [27,28,29,30,31,32]. However, given its brevity and broad conceptualization of health literacy [26], the HLSAC may be a valuable health literacy measuring instrument among adults as well. Moreover, there could be significant value in having two analogous instruments for evaluating health literacy among adolescents and adults when assessing health literacy across diverse age groups and within families.

To our knowledge, the HLSAC has never been validated among adults nor with Rasch model statistics. Rasch analysis is a psychometric technique rooted within Item Response Theory (IRT), and it offers a framework for assessing the dimensionality, reliability, and validity of health literacy measures, thereby informing the development of more precise and accurate measures of health literacy [33].

This study aimed to adapt the HLSAC questionnaire to an adult population through assessment of content validity and subsequently determine the structural validity of the adapted instrument, the Brief Health Literacy scale for Adults (B-HLA), among adults.

## 2. Materials and Methods

### 2.1. Health Literacy for School-Aged Children (HLSAC)

The HLSAC questionnaire is a 10-item self-report health literacy measuring instrument, which is based on a reflective model, i.e., the construct is reflected by the items, in contrast to a formative model, where the construct is the result of the items [34]. Respondents are asked to indicate their response on a 4-point scale with the following response categories: Not at all true (1 point); Not completely true (2 points); Somewhat true (3 points); Absolutely true (4 points). The sum score indicates the respondent’s health literacy level: low health literacy (10–25 points), moderate health literacy (26–35 points), and high health literacy (36–40 points) [35]. Based on a three-phase validation study, which included a standardized forward–backward translation process, interview-based item adaptation, and assessment of structural validity, the Danish HLSAC version has been deemed a reliable and valid health literacy measuring instrument among children aged 11 to 14 years [30].

### 2.2. Study Design

This validation study was carried out in two phases. First, the content validity of the original HLSAC instrument was assessed. This evaluation involved cognitive interviews with adults and group discussions with experts, focusing on item relevance, comprehensiveness, and comprehensibility [36]. Based on this, the instrument was adapted to fit an adult population, thereby establishing the initial version of the Brief Health Literacy scale for Adults (B-HLA) (phase one). Second, the structural validity of the adapted instrument (B-HLA) was tested using Rasch analysis (phase two). The methodologies applied in the two phases are described in detail in the following sections. It should be noted that we refer to HLSAC, i.e., ‘HLSAC score’, pertaining to results from phase one, and to B-HLA, i.e., ‘B-HLA score’, regarding phase two results, in order to reflect the instruments utilized in the respective phases.

### 2.3. Phase One: Content Validity

In September 2022, experts, including two researchers with extensive knowledge about health literacy and health literacy measuring instruments, engaged in a group discussion aiming to investigate relevance, comprehensiveness, and compressibility of the instructing text, items, and response categories. Notes were taken by another researcher during the discussion.

Individual cognitive interviews [37] were performed by one researcher in October 2022 among adults visiting their general practitioner. The researcher had no affiliation with the clinic or the patients. The interviews were carried out in a clinic located in a rural area in Denmark; this clinic was selected for its diverse patient population. Informants were approached in the waiting room and initially received oral and written information about the study before giving consent to participate. 

The purpose of the cognitive interviews was to understand the informants’ cognitive processes and identify any challenges they might encounter when responding to the questionnaire. This approach aimed to examine the relevance, comprehensiveness, and comprehensibility of each individual item [36]. Informants were instructed to think aloud, i.e., verbalize their thoughts and considerations while answering the HLSAC questionnaire [37]. When necessary, predefined probes from our interview guide were used to guide informants to ensure information on relevance, comprehensiveness, and comprehensibility of both the instructing text, items, and response categories [38]. Notes were taken during all interviews. Informants’ sex, age, and HLSAC score were monitored to ensure diversity across the included informants.

Descriptive analyses were performed using the statistical software package R [39]. Notes from cognitive interviews were analyzed to explore the informants’ perceptions of item relevance, comprehensiveness, and comprehensibility, as well as of the instructing text and response categories.

Results from the experts’ discussion and from the cognitive interviews were discussed in the research group, which included the author of the original HLSAC questionnaire (OP). The wording of the questionnaire was revised in accordance with the analysis and the discussions, thereby establishing the initial version of B-HLA scale. Similar to the HLSAC, the B-HLA is a 10-item instrument for measuring health literacy across five core components: theoretical knowledge, practical knowledge, critical thinking, self-awareness, and citizenship (with two items allocated to each component). Respondents express their responses on a 4-point scale using the following categories: Not at all true (1 point), Not completely true (2 points), Somewhat true (3 points), and Absolutely true (4 points). The cumulative score is calculated, and cut-off values regarding respondent’s health literacy from the HLSAC are applied in the initial version of B-HLA: low health literacy (10–25 points), moderate health literacy (26–35 points), and high health literacy (36–40 points).

### 2.4. Phase Two: Structural Validity

In April 2023, adults (aged 18+ years) visiting a general practitioner were approached individually in the waiting room of two clinics covering rural and urban areas in Denmark. The potential respondents were informed orally and in writing about the study. If consenting to participate, respondents completed the B-HLA scale on paper or online by using a QR code. Four questions on age, sex, educational level, and native language were also included in the questionnaire.

The characteristics of the respondents were described through descriptive analyses performed in the statistical software package R [39]. Rasch analysis was performed with Rasch Unidimensional Measurement Models (RUMM2030) software [40].

The Rasch model operates under the assumption that the likelihood of a particular response depends solely on two factors: (1) the difficulty level of the item and (2) the respondent’s ability to address the item. Consequently, if the observed data fit the Rasch model, both items and persons can be positioned on the same scale, essentially providing an objective measure. Rasch analysis evaluates whether the data reflect unidimensionality of the latent trait, local independence among items, absence of differential item functioning, and monotonicity in the expected item score [41]. Hence, the Rasch model meets the requirements of construct validity. When data fit the Rasch model, there is no necessity to explore alternative models for explaining the data, and the measuring instrument functions as a scale that can be used to discriminate across health literacy levels.

Since HLSAC has a polytomous structure (more than two response categories), we initially conducted a likelihood ratio test to identify the most appropriate version of the Rasch model (unrestricted partial credit model or restricted rating scale model) [42,43]. The following Rasch model measurements were examined: overall fit to the model; individual person fit and item fit; adequacy of response categories; differential item functioning (DIF); local dependency; and inspection of the person–item map, reliability, and dimensionality.

The overall fit of the model was examined with a chi-square probability test on the null hypothesis that the data fits the Rasch model. A non-significant value (*p* > 0.05) indicates a fit to the Rasch model. While summary fit residuals for items and persons do not directly identify a specific misfit at the individual item or person level, they can provide a general sense of how well the data fits the model. A perfect fit would yield an average value of 0 and a standard deviation of 1 [44]. Standardized fit residuals and Bonferroni-corrected chi-square probability values were used to assess individual person and item fit. Good fit is indicated by fit residuals <±2.5 and non-significant chi-square values (*p* > 0.01, Bonferroni-corrected). We also calculated the proportion of respondents with divergent answers, i.e., fit residuals ≥±2.5. The adequacy of response categories was evaluated through analyses of an item threshold map. Ordered thresholds indicate that all response categories have a point along the latent variable at which it is the most likely category to be chosen, i.e., no response category is redundant [45].

Differential item functioning (DIF) was examined for each item concerning sex (male/female), age (dichotomized at the median of 44 years), highest completed education (dichotomized at the level of high school), and native language (Danish/other). DIF was assessed with analysis of variance with Bonferroni corrections. We calculated residual correlations to test for local dependency, assuming no indication of local dependency when values were <0.2 [46]. Through assessment of the person–item threshold distribution map, we examined whether the mean location score for persons was close to zero as an indication of a well-targeted scale [44,47]. The person separation reliability index (PSI) tested the internal consistency of the scale and the discrimination abilities of the B-HLA. Reliability is deemed acceptable when the PSI is >0.7 for groups and >0.85 for individuals [47]. A principal component analysis (PCA) of the residuals was used to assess the dimensionality of the instrument. The PCA identifies the most positive and the most negative factor-loading items on the first component. Subsequently, the scores on these two subsets of items for each respondent were compared using a *t*-test. The scale was considered unidimensional if less than 5% of the *t*-tests were significant.

## 3. Results

### 3.1. Results of Phase One: Content Validity

Experts agreed that the HLSAC possessed high face validity. Discussions among the two health literacy experts highlighted challenges related to the comprehensibility of item 2 since it refers to people and locations simultaneously and is lengthy. Additionally, the term ‘instructions’ applied in item 4 was perceived inappropriate when targeting an adult population, and concerns were raised about the relevance of nature as a theme in item 6. The instructing text was perceived as unnecessarily lengthy, leading to the recommendation to align it with the original Finish version of HLSAC, thereby omitting the parentheses (“only one checkmark per question”).

Respondents of the cognitive interviews (*n* = 25) were aged 18–88 years and represented both sexes (40% male, 60% female). The HLSAC scores spanned all health literacy levels: low health literacy (12%), moderate health literacy (56%), and high health literacy (32%). This is comparable to previous findings in children and adolescents reported by the author of the original HLSAC instrument [35]. No respondents placed more than one checkmark per question.

Respondents considered the instructing text and the response categories relevant, comprehensive, and comprehensible. An analysis of results from the experts’ discussion and from the cognitive interviews revealed minor challenges regarding items 2, 4, 6, 9, and 10. These challenges and the subsequent linguistic changes made are described below and presented in the Appendix A. 

**Item 2** (‘Ability to give ideas on how to improve health in one’s immediate surroundings’): Some respondents found it challenging that this rather long item refers to both individuals and locations simultaneously. Despite this, all respondents perceived the item as intended, i.e., having a broad understanding of the immediate surroundings. Consequently, we decided not to alter the wording, except for changing to the definite form by adding a suffix at the end of the term ‘health’ (sundhed[en]).

**Item 4** (‘Ability to follow instructions given by health care personnel’): Respondents found the word ‘instructions’ inappropriate since adults are more often given advice on how to handle health care problems. Therefore, we changed the word ‘instructions’ (instruktioner) to ‘advice’ (anvisninger).

**Item 6** (‘Ability to judge how one’s own actions affect the surrounding natural environment’): Some respondents perceived ‘surrounding environment’ as the natural biological environment, while others perceived it as “the mental environment around me” or “whether people in my social circle drink or smoke”. Since the original item concerns the natural biological environment, we replaced the word ‘environment’ (miljøet) with the word ‘nature’ (naturen). Respondents who perceived the item to address the biological nature indicated that the item distinguished itself from the rest of the items by not being strongly related to health.

**Item 9** (‘Ability to figure out if health-related information is right or wrong’): It was pointed out that ‘figure out’ (finde ud af) is a rather childlike phrase, and the Danish word for ‘assess’ (vurdere) was considered more appropriate. 

**Item 10** (‘Ability to justify one’s own choices regarding health’): It was highlighted that the Danish term for ‘give reasons for’ (begrunde) indicates that respondents should be able to justify their health behaviors. Since people are not always able to provide justification for, e.g., unhealthy behaviors, the Danish word for ‘explain’ (forklare) was found to be more appropriate and more in line with the original English wording. 

Besides altering the above-mentioned words, the order of words in the instructing text and in items 2 and 7 (‘Ability to find health-related information that is easy to understand’) was changed to standardize all items so that all 10 items started with ‘I can’ or ‘I have’, whereas ‘when necessary’ (items 2 and 7) was left out.

### 3.2. Results of Phase Two: Structural Validity

A total of 326 participants were invited to complete the B-HLA questionnaire, and 24 declined to participate. This corresponded to a response rate of 93%. Two respondents completed only part of the questionnaire. One-fifth of the participants (64 (21%)) responded on paper, and the rest responded online. The respondents’ median age was 44 (18–88) years. Data on respondents’ sex, age, highest completed education, native language, and B-HLA score is presented in Table 1. 

Respondents with incomplete responses (*n* = 2) were excluded from further analysis. Respondents with extreme scores (*n* = 10, B-HLA score = 40) were automatically removed from the fit analysis by the RUMM2030 software since extreme scores do not contribute to the Rasch analysis [40]. A total of 290 responses were included in the analyses. A significant likelihood ratio test (*p* < 0.001) supported the use of the partial credit model.

The overall fit analysis illustrated a misfit to the Rasch model (X^2^ = 59.08, df = 40, *p* = 0.03) (Table 2). The summary fit residual SDs for persons (SD = 1.05) and for items (SD = 1.15) were within the acceptable limits. 

All 10 items had fit residuals <±2.5, and nine items had insignificant X^2^ probability values, which indicated a good fit to the Rasch model. Item 8 had the poorest fit to the Rasch model, followed by item 6 (Table 3). Item Characteristic Curves (ICC) for items 6 and 8 are presented in the Appendix A (Appendix A).

The item threshold map showed ordered thresholds for all 10 items. All four response categories had a point on the latent continuum that was the most likely response option to be chosen (Figure 1). Items 4 and 8 were the easiest items to endorse, while items 2 and 6 were the hardest items to endorse.

We found no DIF for sex, age group, level of education, or native language. Due to the relatively slight variation observed in the mean B-HLA score among different age groups, the age variable was dichotomized at its median. The analysis showed no signs of local dependency among items since no residual correlations were >0.2 above average. The mean location score for persons was 2.26 (SD = 1.39), indicating that the lower part of the scale was used less frequently than the rest of the scale among the respondents in our sample (Figure 2). The PSI was 0.80, indicating that the B-HLA can discriminate across different health literacy levels between groups of respondents.

When discarding item 8 from the analysis, we found an overall fit to the model (X^2^ = 39.52, df = 36, *p* = 0.32) and an individual fit of all items. Similarly, when discarding item 6 from the analysis, we found an overall fit to the model (X^2^ = 37.44, df = 36, *p* = 0.40) and an individual fit of all items, including item 8. The proportion of respondents with divergent answers, i.e., fit residuals ≥ ±2.5, was 3.67% for all items, 3.00% when item 8 was discarded, and 1.00% when item 6 was discarded, which indicates that item 6, especially, possesses certain challenges.

## 4. Discussion

### 4.1. Main Findings

To our knowledge, this is the first study to adapt the HLSAC instrument to fit an adult population and subsequently validate the adapted instrument using Rasch analysis. Our results support the potential use of the initial version of the B-HLA for assessing health literacy among adults. Our evaluation of the content validity of the HLSAC resulted in single-word modifications to five items while the fundamental structure of the instrument was preserved. In our Rasch analysis, we found the B-HLA to be a valid and reliable instrument for assessing self-reported health literacy among Danish adults. However, our analyses indicate that items 6 and 8 possess certain challenges. 

### 4.2. Interpretation of Results

The evaluation of content validity suggested that adults and children/adolescents perceive some words and phrases differently, and we recognized the need for making minor adjustments to certain items. The encountered problems regarded primarily specific words (comprehensibility), while item 6 also presented challenges related to its relevance. Bonde et al. described some issues encountered during the translation of the HLSAC into Danish regarding the choice of terminology for ‘surrounding natural environment’ in item 6 [30]. Yet, subsequent focus group interviews conducted to assess the pupils’ understanding of the item did not reveal any challenges with item 6, which supports potential age-related variance in item interpretation.

Only a few health literacy measuring instruments have been validated with Rasch analysis [12,48,49,50,51,52,53]. Most studies have targeted a specific population, e.g., stroke survivors [48] or patients with type 2 diabetes [49,51]. One study compared three short versions of the commonly used 47-item European Health Literacy Survey Questionnaire (HLS-EU-Q47): the HLS-Q12, the HLS-EU-Q16, and the HL-SF12 [52]. None of the instruments were found to be sufficiently unidimensional when evaluated with partial component analysis. The HLS-Q12 showed the best psychometric properties and was recommended by the authors for future use. The HLS-EU-Q16 had several problems, including multidimensionality, item misfit, DIF, and unordered response categories. The HL-SF12 had problems with unordered response categories, and DIF was reported for several items. Our analyses suggest that the B-HLA can be widely used as a unidimensional instrument. This finding is in accordance with previous research validating the original Finnish HLSAC instrument and the Danish version, as both have been found to fit a one-factor model [25,30]. Since the calculation and interpretation of sum scores are only reasonable when the applied instrument is considered unidimensional, the B-HLA may, according to our results, be a valuable health literacy measurement tool. 

Our Rasch analysis identified items 2 and 6 as the two hardest items to endorse. Both items belong under the core component of citizenship. As defined by Paakkari et al. in the conceptual framework for the HLSAC, citizenship refers to “the ability to act in an ethically responsible way and take social responsibility” [26]. Thus, these items require respondents to consider health issues that go beyond their own health perspective. This complexity may be reflected in the higher item locations, which illustrate that respondents must possess more of the underlying trait (health literacy) to endorse these two items. When content validity was assessed, respondents expressed some comprehensibility issues with item 2 and some challenges regarding the relevance and comprehensibility of item 6. According to our Rasch analysis, item 2 functions well and fits the Rasch model despite its comprehensibility issues. However, item 6 (together with item 8) shows the least ideal individual fit. One possible explanation could be cultural differences. In a Danish setting, it is not common to include ‘the effect of one’s actions on the natural environment’ in the general health perception. In our analysis of the B-HLA, discarding item 6 reduced the percentage of respondents with divergent responses significantly, while less so when removing item 8. Noteworthily, respondents did not express any difficulties with item 8 during the content validity evaluation. Furthermore, when discarding either item 6 or item 8, we obtained both a good overall fit to the Rasch model and a good fit of all individual items. Consequently, future research should explore the possible improvement of the B-HLA by refining (or removing) item 6, which seems to constitute the biggest challenge. 

In our sample, only 6% of respondents had low health literacy, according to the B-HLA. Since the instrument has never been used among adults, no previous studies can validate this finding. Application of the HLSAC among school-aged children has shown that the proportion of respondents with low health literacy varies from 6% to almost 19% across countries [27,32,35,54]. Previous research on almost 16,000 Danish adults found 8% to have inadequate health literacy as measured by the HLS-EU-16 [5]. Yet, comparison with these results is problematic. First, the HLS-EU-16 is not sufficiently unidimensional. Second, a comparison between the HLSAC and the HLS-EU-16 has shown weak convergent validity and no concordance between their classifications of health literacy level [55]. 

The relatively low frequency of low health literacy among our respondents compared to former studies of the HLSAC may partly be explained by higher respondent age. On average, this population is expected to have a higher health literacy level and, also, possibly more confidence in their knowledge and competencies than school-aged children and adolescents. Another explanation could be our choice of setting. Our respondents were adults waiting to see their general practitioner, i.e., people who had already managed to schedule an appointment about a health-related issue and thus demonstrated a basic level of health literacy. Additionally, the item proving easiest to answer was item 4: ‘Ability to follow directions given by health care personnel’. Considering that many of our respondents were probably in the clinic to request such directions, this particular item may have generated lower scores in a different setting. Still, Danish general practice offers free (tax-funded) open access to all residents and handles all levels of care, i.e., health promotion, preventive care, and treatment. Thus, even though our setting might have affected the number of respondents with low B-HLA scores, it may also have helped ensure the diversity found in sex, age, educational level, and health literacy. Furthermore, choosing a proactive recruitment method in the clinic waiting-room setting is likely to have helped ensure the high level of diversity and the high response rate in our sample. 

### 4.3. Perspectives in Research and Clinical Contexts

To the best of our knowledge, the HLSAC has never been evaluated as a tool for rapid clinical assessment of health literacy. Such “screening” procedures may fail to acknowledge the finer details of people’s health literacy, thereby giving rise to neglect of needs and potential stigmatization of people in vulnerable positions. Conversely, if the B-HLA is applicable in such contexts, it may promote the integration of health literacy thinking in clinical practice. Accordingly, future research needs to examine the use of the B-HLA in a clinical setting, including the value of evaluating the five core components individually rather than the sum score, in order to obtain a more detailed assessment of the strengths and limitations of health literacy in patients. 

Future research may also examine the use of the B-HLA in relation to family health. Applying the HLSAC and the B-HLA in combination may provide an effective approach for assessing health literacy across different age groups and within families. This option could also prove useful when exploring the development of health literacy from a life course perspective, i.e., from school age to adulthood. 

Future studies on the B-HLA should aim to include more respondents with low health literacy; this could generate valuable knowledge on the psychometric properties of the lower part of the scale. It is also reasonable to expect that including more respondents with low health literacy would improve the PSI, i.e., the internal consistency of the B-HLA scale. Various measurement instruments have been identified, featuring item quantities ranging from 1 to 80, with only a limited subset employing Item Response Theory to evaluate structural validity. Consequently, caution is warranted when comparing Cronbach’s alpha values, as they may not serve as robust indicators of instrument reliability. Notably, no universally accepted gold standard for assessing health literacy exists [12]. Although instruments fitting to the Rasch model articulate criterion-related construct validity [56], it is important to recognize that this study did not explore the convergent validity of B-HLA. Future studies should evaluate how the B-HLA scores correlate with scores from other validated health literacy measures. Cut-off values for health literacy level applied in the B-HLA are aligned with cut-off values from the HLSAC, but whether these values can be extrapolated to B-HLA needs to be further examined. In addition, future research should aim to validate the B-HLA among different target populations, e.g., people suffering from chronic diseases, since it has been shown that some health literacy measuring instruments work differently among patients with chronic diseases compared to the general population [51,52]. We also believe that it would be valuable to further explore the B-HLA across countries to gain insight into its psychometric properties across different contexts and cultural characteristics. Furthermore, we suggest an investigation of the use of B-HLA in different socioeconomic subpopulations, with a view to adding further evidence to the usefulness of the instrument in research and clinical practice. 

## 5. Conclusions

Based on the assessment of content validity, we modified the wording of the HLSAC slightly. This resulted in the creation of the adapted instrument, the B-HLA. Our findings suggest that the initial version of the B-HLA is a valid and reliable instrument for assessing self-reported health literacy among Danish adults. However, the instrument would benefit from further refinement of specific items to ensure that they comply with the culturally defined health perceptions. This should be followed by a new assessment of content validity, structural validity, and convergent validity. Our results point to the usefulness of the B-HLA, alone or combined with the HLSAC, to measure health literacy in future population research on adults and families. Possibly, the instrument could also serve as a short instrument for measuring health literacy in clinical practice.

## Figures and Tables

**Figure 1 ijerph-20-07071-f001:**
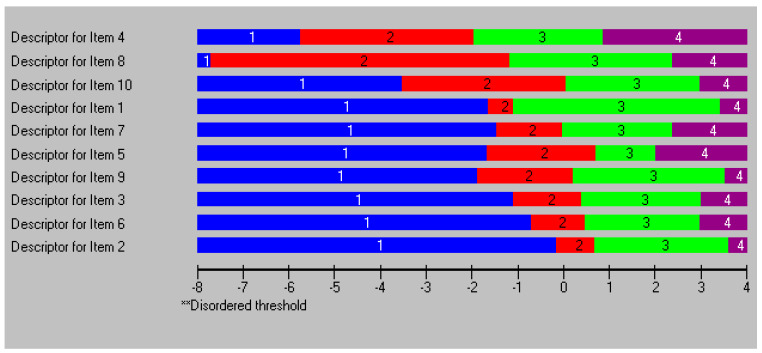
Item threshold map. All 10 items of the Brief Health Literacy scale for Adults (B-HLA) displaying ordered four-point response categories. All items are sorted in location order. 1 (blue): Not at all true; 2 (red): Not completely true; 3 (green): Somewhat true; 4 (purple): Absolutely true.

**Figure 2 ijerph-20-07071-f002:**
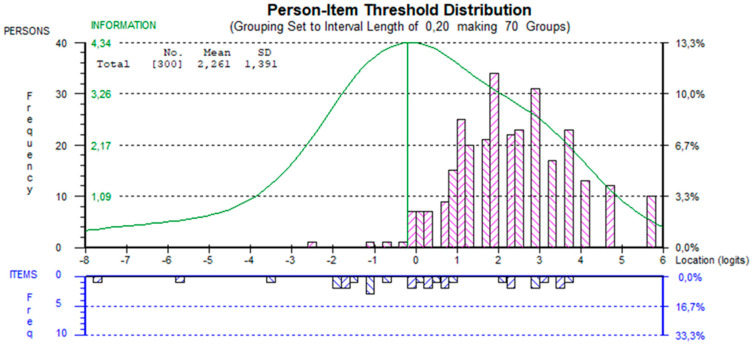
Person–item threshold distribution map.

**Table 1 ijerph-20-07071-t001:** Characteristics of respondents of the Brief Health Literacy scale for Adults (B-HLA) (*n* = 300).

	Observations	B-HLA Score Mean (SD)
	Number (%)	
Sex		
Male	117 (39)	31.5 (4.48)
Female	183 (61)	33.1 (3.93)
Age (years)		
18–29	90 (30)	32.6 (4.31)
30–39	51 (17)	33.3 (3.66)
40–49	25 (8)	31.4 (5.62)
50–59	33 (11)	32.8 (3.70)
60–69	48 (16)	33.1 (3.95)
70–79	38 (13)	31.8 (4.31)
80–89	15 (5)	29.9 (3.48)
Highest completed education		
Not completed primary school	6 (2)	32.3 (3.44)
Primary school	42 (14)	30.6 (4.16)
High school	79 (26)	32.3 (4.22)
Short higher education	54 (18)	31.8 (4.26)
Bachelor’s degree	78 (26)	33.6 (3.95)
Master’s degree	39 (13)	33.6 (3.93)
PhD or doctoral degree	2 (1)	39.5 (0.71)
Native language		
Danish	287 (96)	32.4 (4.22)
Other	13 (4)	34.8 (3.81)
Health literacy level *		
Low (10–25 points)	18 (6)	
Moderate (26–35 points)	207 (69)	
High (36–40 points)	75 (25)	

SD: standard deviation. * Cut-off values for health literacy level are aligned with cut-off values applied in the Health Literacy for School-Aged Children (HLSAC), but whether these values can be extrapolated to B-HLA needs to be examined further.

**Table 2 ijerph-20-07071-t002:** Overall fit statistics for the Brief Health Literacy scale for Adults (B-HLA).

	Overall Model Fit	Item Fit Residual Mean (SD)	Person Fit Residual Mean (SD)	PSI	Unidimensionality Significant *t*-Test (%)
Original sample (*n* = 290)	X^2^ (40) = 59.08, *p* = 0.03	0.09 (1.15)	−0.25 (1.05)	0.80	4.48
Item 8 discarded (*n* = 290)	X^2^ (36) = 39.52, *p* = 0.33	0.11 (1.16)	−0.27 (1.05)	0.77	3.77
Item 6 discarded (*n* = 290)	X^2^ (36) = 37.44, *p* = 0.40	0.13 (1.06)	−0.23 (0.99)	0.78	4.17

X^2^: chi-square; p: probability; SD: standard deviation; PSI: person separation index.

**Table 3 ijerph-20-07071-t003:** Fit statistics for individual items of the Brief Health Literacy scale for Adults (B-HLA).

Item	Response Category Number (%) ^a^				Logit Location	s.e.	Fit Residual	X^2^	d.f.	X^2^ Probability ^b^
	0	1	2	3						
1	2 (1)	16 (6)	195 (67)	77 (27)	0.24	0.12	−0.40	2.47	4	0.065
2	15 (5)	54 (19)	161 (56)	60 (21)	1.38	0.092	0.60	3.86	4	0.043
3	7 (2)	44 (15)	151 (52)	88 (30)	0.77	0.097	−0.08	6.71	4	0.015
4	0 (0)	4 (1)	73 (25)	213 (73)	−2.28	0.14	2.19	7.63	4	0.011
5	4 (1)	46 (16)	106 (37)	134 (46)	0.36	0.094	0.50	5.52	4	0.024
6	9 (3)	45 (16)	148 (51)	88 (30)	0.92	0.094	1.53	9.01	4	0.006
7	4 (1)	31 (11)	132 (46)	123 (42)	0.30	0.10	−1.54	6.63	4	0.016
8	0 (0)	14 (5)	147 (51)	129 (44)	−2.16	0.12	−1.25	14.01	4	0.001
9	4 (1)	42 (14)	177 (61)	67 (23)	0.63	0.11	−0.21	1.58	4	0.081
10	1 (0)	36 (12)	159 (55)	94 (32)	−0.16	0.11	−0.44	1.64	4	0.080

^a^ Response categories: 0: Not at all true; 1: Not completely true; 2: Somewhat true; 3: Absolutely true. ^b^ Bonferroni-corrected chi-square probability; s.e.: standard error; d.f.: degrees of freedom.

## Data Availability

The data that support the findings of this study is available from the corresponding author, S.E.R., upon reasonable request.

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
