# Peer review of "The Brief Health Literacy Scale for Adults: Adaptation and Validation of the Health Literacy for School-Aged Children Questionnaire"

_ijerph, 2023, doi:10.3390/ijerph20227071_

Round 1
Reviewer 1 Report
Comments and Suggestions for Authors
In this study, authors adapted a validated measure for children, i.e., HLSAC instrument, for adults, and subsequently validated the adapted instrument using Rasch analysis.
A big concern from me regarding the study is that some key psychometrics of the new measure were not tested. This can include: 1) how reliable the measurement results from this new measure are, compared to those from a previously validated measure in adult health literacy (‘’the gold standard measure’)? 2) can the results from this new measure correctly reflect the difference between people with different levels of health literacy? It’s inappropriate to simply estimate the levels of health literacy using their scores on HLASC. 3) the convergent validity, i.e., is the score from this new measure positively/negatively correlated with the score from a validated measure on relevant components?
Without these results, it would be insufficient to assess if the new measure is valid or reliable for use.
The English writing is overall easy to follow, with well organised structure.
Below are my comments for authors’ consideration:
Introduction
1. The key components of health literacy can vary dramatically across different age groups of people, including between children and adults. It’s unclear why “HLSAC may be a valuable health literacy measuring instrument also among adults”. It’s inadequate just to mention that “Given its brevity and broad conceptualization of health literacy[31],”.
2. Please provide information on the existing health literacy scales/measures for adults, and explain why it’s necessary and appropriate to adapt MLSAC for adults as a new measure.
Methods
3. Some typologies used in this study can be less known among general readers of the journal. A certain extent of explanations to these terms would be helpful. For example, what did it mean by ‘cognitive interviews’? This similarly applied to ‘Rasch analysis’ and its relevant indices. A brief introduction of these term would be helpful for better comprehension.
4. It is unclear specifically, how the HLSAC was adapted into B-HLA. Please describe this process with details.
5. Please describe the initial structure and contents of the B-HLA scale, following introduction of HLSAC
6. Between section 2.2 and 2.3, some transition sentences are needed.
7. Regarding the informed consent: the interviews were conducted in clinics, it’s necessary to include the role of the interviewer to exclude any possibility of conflict of interests from an ethical perspective.
8. In the descriptions of the two phases across Methods and Resutls sections, ‘The HLSAC scores’ was frequently mentioned. E.g., Line 104-4, p3: ‘Informants’ sex, age, and HLSAC score were monitored to ensure diversity across the included informants’, do you mean ‘B-HLA score’? similarly, in line 89, p2, ‘First, the HLSAC was tested for …’ here do you mean ‘B-HLA’? please check thoroughly across the MS.
9. Line 112, p3: ‘… thereby establishing the B-HLA questionnaire’, its accurate expression could be ‘establishing the initial version of the B-HLA questionnaire’. Here you can also introduce the structure, items/contents of the measurement.
10. The content validity of a new measurement was only tested among laypeople, but not among professionals/researchers in relevant fields. This can be a major limitation of the study, which should be discussed in the Discussion section of the MS.
Results and Discussions
11. Results showed that 5 out of the ten items, i.e., items 2, 4, 6, 9, and 10 were challenging to be understood. For clearer, easier understanding of the main findings, I would suggest authors summarising the main linguistic changes in the main body of the MS, but move Table 1 as a supplement.
Also, because half of the items were rewording, I don’t think it’s appropriate for the authors to say, as noted in the Discussion section, line 323, p9, ‘Our evaluation of the content validity of the HLSAC resulted in minor modifications in the wording,’.
Similarly, in the abstract, there were no relevant results supporting the sentence of “The B-HLA 22 demonstrated unidimensionality and ability to discriminate across health literacy levels.”
Minor changes:
12. Replace “mother tongue” with “native language”
13. There is format error in Table 2.
14. The authors may consider rephrasing the name of the new ‘’questionnaire’, which in my opinion could be named as a ‘measure’ or ‘’scale’.
Comments on the Quality of English Language
Overall acceptable, but certain words/expression can be difficult to understand. e.g., author may replace “mother tongue” with “native language”.
Reviewer 2 Report
Comments and Suggestions for Authors
The Health Literacy for School-Aged Children Questionnaire (HLSAC) served as the basis for the development of the Brief Health Literacy Questionnaire for Adults (B-HLA), whose psychometric qualities are the main focus of this work. The results discovered and presented will be of interest to health psychology researchers and practitioners.
The manuscript is a helpful endeavor, and the professionally done statistical analysis are presented.
One of the primary concerns regarding the article pertains to the absence of sufficient rationale for employing the questionnaire originally designed for children and adolescents in the context of adult participants. The authors have attempted to provide justification for the suitability of the questionnaire using both qualitative means (cognitive interviewing) and quantitative means. However, the extent to which the procedure is valid and applicable to a clinical sample of adults does not appear to be entirely persuasive at this time. The development process is heavily dependent on the psychometric model and does not incorporate theoretical foundations, such as multidimensional features.
One issue pertaining to the manuscript's structure is the inadequate introduction and contextualization of the topic and methodology inside the brief literature review. There exist multiple systematic reviews in the existing body of literature that provide a comprehensive overview of the theoretical framework and measurement instruments employed in methodologies within adult contexts (e.g., Muhanga & Malungo, 2017; Lui et al., 2018).
In the methodological section, it would be worthwhile to emphasize and justify why the Rasch model was chosen and, for example, what age-matching would be professionally justified for the age assessment of DIF.
While the subdivision of the methodological section into concise sections enhances comprehensibility, it is not recommended to isolate particular phrases or short passages of text within a distinct section.
The discussion has a significantly greater level of complexity compared to the introductory section of the literature. Several elements inside the discussion may potentially be relocated to the introduction, provided they were more precisely defined.
In overall, the paper presents a highly valuable initiative; nonetheless, it would benefit from a more comprehensive examination of the method's validity, extending beyond the limitations of the psychometric model.
Liu, H., Zeng, H., Shen, Y., Zhang, F., Sharma, M., Lai, W., ... & Zhao, Y. (2018). Assessment tools for health literacy among the general population: a systematic review. International journal of environmental research and public health, 15(8), 1711.
Muhanga, M. I., & Malungo, J. R. (2017). The what, why and how of health literacy: a systematic review of literature.
Reviewer 3 Report
Comments and Suggestions for Authors
The submitted manuscript presents a Rasch analysis for the adaptation of the brief health literacy questionnaire for adults (B-HLA) to school-aged children. I found the manuscript well-written and have a few comments.
1. 46 and others: There is frequently no space between words and references. E.g., I read “literacy[10-13]” while it should read as “literacy [10-13]”.
2. 64: “Rasch analysis” is anything but a modern psychometric technique. In contrast, it is a quite dated technique.
3. Please provide a reference for the estimation techniques utilized in the RUMM software.
4. 106: I find it strange in an article read that “Qualitative and descriptive analyses were performed by SR.” This is a statement that must appear in the part “Author Contributions” in the paper. See also 121 and 123 for the same issue.
5. 241ff.: It was said that persons with extreme responses were removed from the analysis because they “would not contribute to the Rasch analysis.” I think this statement is incorrect because, according to my knowledge, the RUMM software uses a limited information method that must not automatically remove persons with extreme scores from the analysis (in contrast to the Winsteps software, which relies on JML estimation).
6. Table 4: The authors only provide a “fit residual” and a test statistic for assessing item fit. Please also report item infit and outfit statistics.
7. Discussion or Introduction: It was unclear to me why there would be advantages in applying the Rasch model instead of other modeling approaches for the questionnaire. Why not apply a factor model using (the intentionally misspecified) normal distribution or a graded response model?
8. Ref. [37]: Write “R Core Team.”
Round 2
Reviewer 1 Report
Comments and Suggestions for Authors
My comments were adequately addressed. No further comments.
Reviewer 2 Report
Comments and Suggestions for Authors
Thanks for the detailed answers and the revision.
Reviewer 3 Report
Comments and Suggestions for Authors
no further comments